# Physical/Chemical Properties and Resorption Behavior of a Newly Developed Ca/P/S-Based Bone Substitute Material

**DOI:** 10.3390/ma13163458

**Published:** 2020-08-05

**Authors:** Bing-Chen Yang, Jing-Wei Lee, Chien-Ping Ju, Jiin-Huey Chern Lin

**Affiliations:** 1Department of Materials Science and Engineering, College of Engineering, National Cheng Kung University, Tainan 70101, Taiwan; N58011308@ncku.edu.tw; 2Division of Plastic and Reconstructive Surgery, National Cheng Kung University Hospital, Department of Surgery, College of Medicine, National Cheng Kung University, Tainan 70403, Taiwan; jwlee@mail.ncku.edu.tw

**Keywords:** Ca-based, bone substitute, resorption, animal study, histology

## Abstract

Properly regulating the resorption rate of a resorbable bone implant has long been a great challenge. This study investigates a series of physical/chemical properties, biocompatibility and the behavior of implant resorption and new bone formation of a newly developed Ca/P/S-based bone substitute material (Ezechbone^®^ Granule CBS-400). Experimental results show that CBS-400 is comprised majorly of HA and CSD, with a Ca/P/S atomic ratio of 54.6/39.2/6.2. After immersion in Hank’s solution for 7 days, the overall morphology, shape and integrity of CBS-400 granules remain similar to that of non-immersed samples without showing apparent collapse or disintegration. With immersion time, the pH value continues to increase to 6.55 after 7 days, and 7.08 after 14 days. Cytotoxicity, intracutaneous reactivity and skin sensitization tests demonstrate the good biocompatibility features of CBS-400. Rabbit implantation/histological observations indicate that the implanted granules are intimately bonded to the surrounding new bone at all times. The implant is not merely a degradable bone substitute, but its resorption and the formation of new cancellous bone proceed at the substantially same pace. After implantation for 12 weeks, about 85% of the implant has been resorbed. The newly-formed cancellous bone ratio quickly increases to >40% at 4 weeks, followed by a bone remodeling process toward normal cancellous bone, wherein the new cancellous bone ratio gradually tapers down to about 30% after 12 weeks.

## 1. Introduction

According to the reports of Weiser et al. [1] and Hall et al. [2], the number of orthopedic surgeries performed worldwide was approximately 24 million in 2004, and forecast to grow to 31 million by 2012. One of the most challenging issues among these surgeries has been the reconstruction of bone defects caused by trauma, tumor removal, congenital deformity, etc., therein, 20–25% cases would require bone grafting [3]. Bone grafting is also highly demanded in dental industry. Institut Straumann AG (Basel, Switzerland) estimated that the total number of dental implants used in 2018 worldwide could be more than 10 million [4]. Cha et al. [5] estimated that bone grafting is required for about one in every four dental implants, meaning that the demand for bone substitutes would be more than 2.5 million units for dental implant applications alone.

Being osteoconductive/osteoinductive, non-immunogenic and able to improve the healing process, autologous bone grafts have been considered the gold standard for the repair of osseous defects [6]. Autologous bone grafts harvested from the iliac crest are commonly used in reconstructive orthopedic surgeries. However, considerable morbidity has been reported related to iliac crest harvesting, with a high complication rate up to 49%, including damage to blood vessels and nerves, joint disruption, fractures, subluxation, herniation of abdominal contents, and delayed iliac abscess [7]. Another major concern with the iliac crest harvest procedure is that the amount of bone available for autografting is limited [8], which has prompted an increased interest in using bone graft substitutes [6].

Xenografts, allografts and synthetic bone substitutes are some common alternatives to autologous bone. However, diseases carried from human or animal bone-derived grafts always pose a potential risk to the patients receiving such grafts. Transmission of human immunodeficiency virus (HIV), hepatitis C virus (HCV), human T-lymphotropic virus (HTLV), unspecified hepatitis, tuberculosis and other bacteria has been documented to be associated with allografts [9]. Xenografts derived from animal bone also carry the risk of transmission of diseases. Kim et al. [10] reported that anorganic bovine bone, a popularly used xenograft in dentistry, has the risk of transmitting bovine spongiform encephalopathy (BSE) prion PrPSc. Proteins were detected in Bio-Oss^®^ (bovine-bone derived) [11] and tibia samples treated under a similar deproteinization condition [12]. More recently, Kim et al. [13] suggested that humans are not safe from the infection of prion disease of other species, due to its ability to cross the species barrier. These findings indicate that the long-term risks of bovine bone-derived xenografts, which are extensively used in dentistry today, should be seriously considered.

Apparently, the use of synthetic materials as bone grafts can avoid the aforementioned disease transmission risks. An ideal degradable/resorbable bone substitute material should have a degradation/resorption rate comparable to the rate of the host repairing bone to facilitate a complete bone repair [14]. Nevertheless, most of the currently available bone substitute materials resorb either too fast or too slowly [15]. For example, hydroxyapatite (HA), a widely-used calcium phosphate, generally resorbs too slowly, while calcium sulfate, another popularly-used resorbable bone substitute, is often considered to resorb too fast [16]. It was reported that an implant resorbing too slow may hamper bone repair, degrade mechanical properties of the repaired bone, and lead to chronic inflammation [15,17]. On the other hand, an implant resorbing too fast can lead to resorption without sufficient new bone ingrowth [18]. There is no doubt that properly regulating the resorption rate of a resorbable bone implant remains one of the biggest challenges in this field.

To avoid disease transmission risks, optimize resorption rate and osteoconductivity, a synthetic, inorganic and highly porous Ca/P/S-based bone-substituting material (Ezechbone^®^ Granule CBS-400) has been newly developed by a National Cheng-Kung University (NCKU)/ Joy Medical Devices (JMD) joint research project. The purpose of the present study was to investigate a series of physical/chemical properties, biocompatibility, and particularly the behavior of implant resorption and new bone formation of this material in a rabbit implantation study.

## 2. Materials and Methods

The present Ca/P/S-based material for the study (Ezechbone^®^ Granule CBS-400) was proprietarily manufactured by JMD, an ISO 13485/GMP-certified facility in Kaohsiung, Taiwan. The raw materials used to fabricate CBS-400 were all purchased from accredited manufacturers with US Pharmaceutical (USP) grade. The heavy metal (primarily As, Cd, Hg and Pb) concentrations of the final product determined by inductively coupled plasma-mass spectrometry (ICP-MS) (Element XR™, Thermo Fisher Scientific Inc., Waltham, MA, USA) were all within the cited limitations as set forth in ASTM F1185-03 Sections 4.3 and 4.4. To assess the safety and efficacy of the present bone-substituting material, a series of chemical/physical characterization and biocompatibility tests were conducted. The physical/chemical testing items of the present study included phase identification, Ca/P/S atomic ratio, granule morphology, dimensional stability, porosity volume fraction, pore size, pH value, and solubility tests in buffered citric acid and TRIS-HCl solution. The biocompatibility tests included cytotoxicity, intracutaneous reactivity, skin sensitization and animal implantation tests.

### 2.1. Phase Identification and Chemical Composition

An X-ray diffraction (XRD) system (D2 PHASER, Bruker, Billerica, MA, USA) was used to scan the samples in the range from 5° to 55° (2 theta), at a scan speed of 1°/min. The XRD patterns were obtained using a Ni-filtered Cu-Kα radiation diffractometer, operated at 30 kV and 10 mA equipped with a diffracted-beam monochromator. For better resolution, the Kα2 signal was stripped and only Kα1 X-ray with a wavelength of 1.54060 Å was used for analysis. The XRD patterns were analyzed by matching each characteristic peak with that compiled in the ICDD/JCPDS cards of hydroxyapatite (HA), calcium sulfate dihydrate (CSD), tetracalcium phosphate (TTCP) and anhydrous dicalcium phosphate (DCPA).

The Ca/P and Ca/P/S ratios of the CBS-400 granules were determined using a scanning electron microscope (SEM) (JSM-6510, JEOL, Akishima, Japan), equipped with an energy dispersive spectrometer (EDS) (X-ACT, Oxford Instruments, Abingdon, UK) operated at 10 kV and 70 mA, with a working distance of 10 mm. The granules were ground into powder and then packed into a 6 mm-diameter cylindrical stainless steel mold under a pressure of 100 kgf to form a cylindrical disc. To avoid electric charging of the non-conducting material, the disc samples were sputter-coated with a thin layer of carbon, using a JEOL JEC-560 AUTO CARBON COATER system.

Prior to a semi-quantitative analysis of the samples, an overall chemical analysis was conducted on large areas. The results indicated that the predominant elements of CBS-400 were Ca, P, S, O and C, wherein C signals primarily came from the sputtering process and O signals were also commonly detected due to the unavoidable contamination from the environment. To exclude these artifact effects, the Ca/P and Ca/P/S ratios were normalized by assuming Ca + P + S = 100%.

### 2.2. Granule Morphology and Dimensional Stability in Hank’s Solution

The granule morphology and dimensional stability of the material immersed in Hank’s solution were evaluated using the same JEOL JSM-6510 SEM. Three granular samples were immersed in daily refreshed Hank’s solution at 37 °C, with a pH value of 7.4 and a sample/liquid ratio of 1 g/20 mL. One sample was taken out after 1 day; the second sample after 3 days; and the third after 7 days. The immersed granules were dried in an oven at 50 °C for 24 h, then mounted on a specimen stub using a carbon tape. To avoid electric charging during SEM examination, the non-conducting samples were sputter-coated with gold for 120 s at 20 mA, using a vacuum sputter system (E-1045, Hitachi High-Technologies Corp., Tokyo, Japan).

To determine particle size distribution of the granules, the CBS-400 particles were randomly placed on a transparent glass slit with a black matte finish paper underneath, and photographed using a digital camera (D90 DSLR, Nikon, Tokyo, Japan). Particle sizes (Minimum Feret diameter) were determined using ImageJ (ver.1.52a) issued by the National Institutes of Health (NIH). Through the Color Threshold function, the plain-white particles were selected from the matte black background. Using the Convert to Mask function, a binary image with scattered black particles was obtained. The Open function, a binary processing tool, was applied to largely eliminate the noise from photographing. The Watershed function, another binary processing tool, was performed to avoid occasionally aggregated particles. The binary image-processed figures were then analyzed using the Analyze Particles function, generating the needed distinct particles max and min Feret diameter information.

### 2.3. Porosity Volume Fraction and Pore Size

The porosity value of the granular product was determined according to ASTM C-830-00. The measurement was conducted using the liquid (95% v/v ethanol) intrusion method, wherein the apparent porosity, P, was determined by the equation, P (% v/v) = [(saturated weight-dry weight)/(saturated weight-suspended weight)] × 100%.

The pore sizes of the granules were determined using the same SEM and analyzed through ImageJ (ver.1.49). Then, 3 lots of CBS-400 were randomly selected and tested. One sample was randomly selected from each lot and 10 sites were randomly selected from each sample. For each site, 50 pores were randomly selected for measurement. In so doing, a total of 500 pores were measured for each lot and its pore size distribution was obtained.

### 2.4. pH Value

The pH values of the daily refreshed Hank’s solution at 37 °C with a pH value of 7.4, wherein CBS-400 was immersed with a sample/liquid ratio of 1 g/20 mL for 1, 2, 3, 4, 5, 6, 7 and 14 days, were measured. A pH meter (SP-2300, Suntex Instruments, Taipei, Taiwan) was used for the test. The average pH values at each time point and their changes with immersion time were analyzed.

### 2.5. Solubility Testing in Buffered Citric Acid and TRIS-HCl Solution

The concentrations of Ca, P and S elements released from CBS-400 immersed in citric acid solution and simulated body fluid were measured and compared to the concentrations of these elements in human blood. According to ISO 10993-14:2001(E), this test consists of two parts: the extreme solution test, wherein a low pH buffered citric acid solution with a pH value of 3 at 37 °C was used as a worst-case low-end service environment, and the simulation solution test, wherein a TRIS-HCl buffer solution with a pH value of 7.4 was used to simulate the body’s normal pH level.

For the extreme solution test, four CBS-400 samples were immersed for 5 days in the freshly-prepared buffered citric acid solution, which was agitated longitudinally at 2 Hz at 37 °C, with a sample/liquid ratio of 1 g/20 mL. After 5 days, the samples were removed from the containers via filtration and the filtrate was retained for analysis. The concentrations of Ca, P and S elements in the filtrate and blank buffered citric acid solution were analyzed using an ICP-MS system (7500ce, Agilent Technologies, Inc., Santa Clara, CA, USA).

The TRIS-HCl solution for the simulation solution test was prepared by dissolving a desired amount of tris(hydroxymethyl)aminomethane in water. The pH value of the solution was adjusted by HCl to 7.4 at 37 °C. The CBS-400 samples were immersed in the TRIS-HCl solution, which was agitated longitudinally at 2 Hz at 37 °C, with a sample/liquid ratio of 1 g/20 mL for 1, 3 and 5 days. After each time point, 4 samples were removed from the containers via filtration, and the filtrates were retained for analysis. The concentrations of Ca, P and S elements in the filtrates and blank TRIS-HCl solution were analyzed using the same ICP-MS system.

The conversion between concentrations (ppm) and mass of elements released (mg) was based on the equation, [Element concentration in filtrate (ppm) − Element concentration in blank vehicle (ppm)] × Immersion solution volume (mL) = Mass of element released from CBS-400 (mg). The calculated value represented the mass of the element released from 1 g CBS-400, immersed in 20 mL immersion solution at 37 °C.

### 2.6. Biocompatibility

To assess the safety of CBS-400, a series of biocompatibility tests, including cytotoxicity, intracutaneous reactivity and skin sensitization, were conducted. For intracutaneous reactivity and skin sensitization tests, the animal test protocols adopted were reviewed and approved by the Institutional Animal Care and Use Committee of NCKU (IACUC Approval Number: 102254 and 103309).

#### 2.6.1. Cytotoxicity

The cytotoxicity test, which was designed to determine the biological response of mammalian cells in vitro, was performed according to ISO 10993-5:2009(E) methods, wherein the extraction method was used for the study. The extract of CBS-400 in culture medium with serum at a ratio of 0.1 g/mL was used as the test sample. The CBS-400 test samples and the negative control, Al_2_O_3_ particles, were gamma-ray sterilized at a dosage of 25 kGy. Culture medium with serum was used as the blank control. An established cell line (NIH/3T3 cells, BCRC 60008) was used for the cytotoxicity test, due to its sensitivity to chemical-induced cytotoxicity [19].

Cell viability was determined by WST-1 assay, a colorimetric assay for mitochondrial dehydrogenase activity in which the absorbance at 450 nm is proportional to the amount of dehydrogenase activity in the cell [20]. Overall, 6 wells were tested for control groups and 24 wells for CBS-400 extract. According to ISO 10993-5:2009(E), the test material has a cytotoxic potential if its cell viability value is less than 70% that of the blank control.

#### 2.6.2. Intracutaneous Reactivity Test

The potential of CBS-400 to produce irritation was assessed by an intracutaneous reactivity test according to ISO 10993-10:2010(E). Three healthy young adult male New Zealand White (NZW) rabbits weighing >2 kg were used for the test for each positive control and CBS-400 extract group. For the CBS-400 extract group, the extracts were separately prepared by immersing CBS-400 granules in polar (saline) and non-polar (sesame oil) solvents, at a ratio of 0.1 g/mL at 37 °C for 72 h. The arrangement of the injection sites followed that set forth in ISO 10993-10:2010(E) clause 6.4.5. The positive control group followed the same procedure, wherein formalin and histamine were used as polar and non-polar positive controls, respectively. The appearance of each injection site was noted immediately after injection, and at 24, 48 and 72 h after injection. The tissue reactions for erythema and oedema at each injection site and each time interval were graded according to the system described in detail in ISO 10993-10:2010(E), clause 6.4.6.

After the grading at 72 h, all erythema grades and oedema grades at 24, 48 and 72 h were totaled for each CBS-400 extract test sample and blank on each individual animal separately. To calculate the score of a test sample or blank on each individual animal, each of the totals was divided by 15 (3 scoring time points ×5 CBS-400 extract or blank sample injection sites). To determine the overall mean score for each test sample and the corresponding blank, the scores for the three animals were averaged. The final test sample score was obtained by subtracting the score of the blank from that of the test sample. The requirement of the test was met if the final score was 1.0 or less.

#### 2.6.3. Skin Sensitization Test

Skin sensitization was assessed by murine local lymph node assay (LLNA): 2-bromodeoxyuridine-enzyme linked immunosorbent assay (BrdU-ELISA) test according to ISO 10993-10:2010(E), OECD guideline for the testing of chemicals 442B, and a report of Takeyoshi et al. [21]. Overall, 15 healthy, 8–9 week non-pregnant female mice of the CBA/CaJNarl strain were used for the study; 5 mice for each group. The extract of CBS-400 in acetone olive oil (AOO) solution (4:1 v/v) was used as the test sample. According to OECD 442B, AOO solution and 2,4-dinitrochlorobenzene (DNCB) were selected respectively as blank and positive control for this test. The extract was prepared by immersing CBS-400 granules in the solvent, at a ratio of 0.1 g/mL at 37 °C for 72 h.

Following an ISO 10993-10:2010(E)-recommended study of Takeyoshi et al. [21], the control substances or extracts were applied to the dorsum of both ears daily for three consecutive days. A single intraperitoneal injection of BrdU (5 mg per mouse per injection) was given on day 4. Then, 24 h after injection of BrdU, the mice were sacrificed and their auricular lymph nodes were removed, weighed and stored at −20 °C prior to analysis with ELISA to measure the level of BrdU incorporation. Unpaired Student’s t-test was used to analyze the differences in body weight change, lymph nodes weight and labeling index. The incorporation of BrdU into lymph node cells (LNC) was determined using a commercial cell proliferation assay kit (BioVision Inc., Milpitas, CA, USA; Cat. No. K306-200). 

The absorbance at 450 nm was determined using a microplate reader (Multiskan™,Thermo Fisher Scientific Inc., Waltham, MA, USA), with a reference wavelength of 630 nm. The absorbance was defined as the BrdU labeling index. Means and standard errors for the labeling indices were calculated for each treatment group. The stimulation index (SI) was calculated by dividing the labeling indices in each test group by that in the concurrent blank control group. According to the principle set forth in OECD 442B, the response in cellular proliferation of 1.6 times or more compared to the activity of the vehicle control is a threshold for designating a test material as a sensitizer.

### 2.7. Animal Implantation Study and Resorption Rate of CBS-400 Implant

An animal implantation study was conducted to evaluate local tissue responses and the resorption process of the CBS-400 granules in the implantation sites and their adjacent areas after 4, 8 and 12 weeks of implantation in the femur condyle of New Zealand White (NZW) rabbits. The animals were acclimatized and cared for as specified in ISO 10993-2:2006(E). Moreover, 22 adult, healthy, male NZW rabbits weighing 2.5–3.5 kg were used for the study. The rabbits were housed individually in stainless steel cages with free access to food and water. An acclimation period of a minimum of 7 days was allowed between receipt of the animals and the start of the study. The animal study was performed at NCKU Medical College Animal Center (Tainan, Taiwan). The animal test protocols adopted were reviewed and approved by the Institutional Animal Care and Use Committee of NCKU (IACUC Approval Number: 102254).

All the animals were operated on under general anesthesia. Zoletil 50 (0.05 mL/100 g, Virbac, Carros, France) was used as general anesthesia, while xylocaine (AstraZeneca, Cambridge, England, UK) was used as a local anesthesia. The implantation sites were shaved and cleansed with 70% v/v ethanol and Betadine^®^ (povidone iodine 10% w/v, Mundipharma, Frankfurt, Germany). To implant CBS-400 in the medial epicondyle of the femur, a longitudinal incision was made on the anterior surface of the femur. The inner side of the knee joint was cut to expose the femur. After that, the periosteum was reflected, and a 2 mm pilot hole was drilled, which was sequentially widened with drills of increasing size until a final diameter of 5 mm was reached. A 5 mm diameter drill burr was used and a ring was inserted at a depth of 10 mm to ensure the appropriate length (10 mm) of the drill hole. (Figure 1). A cylindrical void of 5 mm in diameter and 10 mm in depth was created in rabbit femoral condyle. Cavities without filling were used as a negative control. After filling, subcutaneous tissues and skin were closed up, layer by layer, with sutures.

The animals were sacrificed at 3 days (3D), 4 weeks (4W), 8 weeks (8W) and 12 weeks (12W) post-operation. The animal numbers were 3 for the 3D group, 1 for the negative control and 6 for the other groups. The harvested samples were divided into two groups: hematoxylin and eosin (H&E)-stained decalcified samples for investigating inflammation/immune responses and toluidine blue (TB)-stained non-decalcified samples, for investigating implant resorption behavior. TB staining was preferably used for the study of the implant resorption behavior, due to its better overall performance in the quantitative determination of implant resorption and new bone formation rates.

After sacrifice, the femur portions were excised immediately and the excess tissues were removed. The retrieved bone was sectioned using a low-speed diamond blade (IsoMet Blade, Buehler, IL, USA). For consistency, the samples for histological examination were taken from the section located between 2 mm and 6 mm from lateral to medial (section B or C shown in Figure 1). For regular H&E staining, the sectioned samples were fixed in 10% w/v neutral buffered formalin (NBF) (pH 7.0) for 3 days, decalcified, dehydrated, and embedded in paraffin. After embedding in paraffin, 5 μm-thick sections of the samples were obtained using a rotary microtome. The slices were then plated, deparaffinized, and stained with H&E. Non-decalcified samples were fixed in the same way, dehydrated in increasing grades of ethanol, then embedded in Buehler EpoxiCure 2 resin. The embedded samples were ground using silicon carbide grit paper, followed by wet-cloth polishing with 1.0, 0.3 and 0.05 μm Al_2_O_3_ powder, then stained with TB. The samples were examined using a light microscope (DM2500P, Leica, Wetzlar, Germany). For the TB-stained non-decalcified samples, in order to enhance resolution and at the same time to have an overall picture of implant resorption behavior, more than one hundred reflected-light micrographs at a magnification of ×100 were taken sequentially on each section being examined. These 100+ micrographs were then superimposed by Leica Application Suite software, to form a large composite picture covering the entire implant/bone cross-section.

To determine the residual implant ratio, the original 5 mm dia. bone void zone was carefully checked, defined, and artificially contoured on the superimposed composite image being examined. The implant residues appearing on these composite images were distinguished manually. The different regions of new bone, residual implant material, and epoxy that filled the empty space (such as bone marrow) were sketched out according to their respective colors, contrasts and histological features. An image-analyzing software, ImageJ (ver.1.51 g), was used for counting pixels of new bone, bone marrow and residual implant material. The area of bone in the healthy condyle of 12W rabbits without surgery was determined by the same method. The residual implant area ratios were determined according to the equations,
Residual implant ratio (%) = Area of residual implant/Area of original artificially-drawn bone void region
and
New bone formation ratio (%) = Area of new bone/Area of original artificially-drawn bone void region.

Due to the fact that the drawing of an original 5 mm dia. bone void circle could not be absolutely accurate, to assure the entire analyzed area was within the original bone void zone, another 4 mm dia. circle was drawn inside the 5 mm dia. circle. The residual implant and new bone ratios within this smaller circle were also measured and compared with the data from the 5 mm dia. circle.

### 2.8. Statistical Analysis

The quantitative data are presented as means ± standard deviations. Box plots were used to demonstrate the 5th percentile, 1st quartile (25th percentile), median (50th percentile), 3rd quartile (75th percentile) and 95th percentile, to show the data distribution. One-way ANOVA was used to evaluate the difference in pH values in Hank’s solution and the masses of Ca, P and S elements released from CBS-400 in TRIS-HCl group between different immersion days. For the skin sensitization test, unpaired Student’s *t*-test was used to analyze the differences in body weight change, lymph nodes weight and labeling index. Unpaired Student’s *t*-test was also used to analyze the differences in the masses of Ca, P and S elements released from CBS-400 between TRIS-HCl group and citric acid group, residual implant and new bone ratios between the two circles, as well as the differences in bone area ratio between blank control and implantation groups. Linear regression and polynomial regression (quadratic regression) analyses were used to evaluate the change of residual implant ratio and new bone formation ratio versus implantation time, respectively. To ensure the calculated areas were always within the implantation region, the smaller, 4 mm-dia. circle was chosen for regression analysis. A statistical significance is determined at *p* < 0.05.

## 3. Results

### 3.1. Phase Identification and Chemical Composition

As shown in Figure 2A,B, CBS-400 is comprised majorly of HA (JCPDS# 09-0432) and calcium sulfate dihydrate (CSD, JCPDS# 33-0311), with relatively small amounts of tetracalcium phosphate (TTCP, JCPDS# 25-1137) and dicalcium phosphate anhydrate (DCPA, JCPDS# 09-0080). The semi-quantitatively-determined average phase contents of CBS-400 are 40.7 wt% HA, 28.6 wt% CSD, 12.5 wt% TTCP and 18.2 wt% DCPA. The SEM-EDS results shown in Figure 2C indicate that the major detected elements of CBS-400 are Ca (Kα = 3.7 and Kβ = 4.0 KeV), P (Kα = 2.0 KeV) and S (Kα = 2.3 KeV), O (Kα = 0.53 KeV) and C (Kα = 0.28 KeV). The normalized (by assuming Ca + P + S = 100%) Ca/P/S and Ca/P atomic ratios of CBS-400 are 54.61/39.21/6.18 and 1.39, respectively. Raw data of phase contents and SEM-EDS results are provided in Appendix A and Appendix A, respectively.

### 3.2. Granule Morphology and Dimensional Stability in Hank’s Solution

The CBS-400 granules designed to have particle sizes ranging from about 400 μm to 1200 μm were sieved between 16 mesh (1.18 mm) and 40 mesh (0.43 mm). It should be remembered that, due to the irregular shape of the particles, certain elongated particles with average particle dimensions larger than the 16-mesh sieve (1.4 mm) could unavoidably pass through the sieve.

As shown in Figure 3A–C, the CBS-400 granules exhibit an irregular, rough and porous morphology with numerous macro pores. Figure 4A shows that the granules have sizes predominantly between 300 µm and 1700 µm. The measurements indicate that less than 5% of the particles are smaller than 300 µm, and about 90% of the particles are between 300 to 1300 µm, close to our design. Raw data of particle sizes are provided in Appendix A.

Figure 3D–L represents the morphology of CBS-400 granules after immersion in Hank’s solution for 1, 3 and 7 days. After immersion for 7 days, the overall morphology, shape and integrity of the granules remain similar to that of the non-immersed samples, without showing an apparent collapse or disintegration under SEM.

### 3.3. Porosity Volume Fraction and Pore Size

The measured average porosity value of the CBS-400 granules, 77.8% v/v, was found to be higher than the theoretical volume fraction of the pore-former (~73% v/v), indicating that the pore-forming particles had been substantially rinsed out during processing. As shown in Figure 4B, the pore sizes of CBS-400 are majorly distributed between about 60 μm and 180 μm, with an average pore size of 112 μm. The distribution data indicate that 98–99% of the measured pores of the granules are between 40 μm and 180 μm; and about 90% between 60 μm and 160 μm. It should be noted that the SEM-measured pore sizes would always be somewhat smaller than the “actual” pore sizes, due to the fact that the largest dimensions of the pores being examined under SEM are probably “hidden” underneath the viewing surface. Raw data of pore sizes and porosity values are provided in Appendix A and Appendix A, respectively.

### 3.4. pH Value

The pH values of the daily refreshed Hank’s solution, wherein CBS-400 was immersed for 1, 2, 3, 4, 5, 6, 7 and 14 days, are given in Figure 5. As seen in the figure, the average pH value of the 1-day solution drops to 5.28 (mildly acidic). With increasing immersion time, the pH value continues to increase significantly with immersion time. The pH value increases to 6.55 after 7 days, and 7.08 after 14 days. Raw data of pH values are provided in Appendix A.

### 3.5. Solubility Tests in Buffered Citric Acid and TRIS-HCl Solutions

The concentrations and calculated masses of Ca, P and S released from CBS-400 in buffered citric acid solution and TRIS-HCl solution are shown in Figure 6. In the extreme solution test, the average concentrations of Ca, P, and S in buffered citric acid solution are 1208.1, 1285.4 and 570.9 ppm, respectively, and their respective calculated masses of Ca, P, and S elements released from CBS-400 are 24.2, 25.7 and 10.2 mg.

In the simulation solution test, after a 1-day immersion, the concentrations of Ca, P, and S in TRIS-HCl are 243.8, 4.6 and 443.6 ppm, respectively, and their respective calculated masses of Ca, P and S elements are 4.9, 0.09 and 8.9 mg. After a 3-day immersion, the concentrations of Ca, P and S in TRIS-HCl are 360.2, 9.4 and 603.3 ppm, respectively, and their respective calculated masses of Ca, P, and S elements are 7.2, 0.19 and 12.1 mg. After a 5-day immersion, the concentrations of Ca, P and S in TRIS-HCl are 413.3, 14.5 and 670.2 ppm, respectively, and their respective calculated masses of Ca, P and S elements are 8.3, 0.29 and 13.4 mg. At day 5, compared to the TRIS-HCl group, the citric acid group shows significant differences in Ca, P and S elements released from CBS-400. Within the TRIS-HCl group, the released Ca, P and S elements significantly increase with immersion time. Raw data of particle sizes are provided in Appendix A.

### 3.6. Biocompatibility

#### 3.6.1. Cytotoxicity

As shown in Figure 7, the NIH/3T3 cells treated with the extract of CBS-400 for 24 h depict adherent and extended morphology (Figure 7C), similar to that treated with medium (Figure 7B) or Al_2_O_3_ extract (Figure 7E), confirming that CBS-400 extract did not inhibit cell growth. On the other hand, when treated with 0.3% phenol, the cell morphology was seen to change to round-shaped (Figure 7D), indicating that the positive control had strong cytotoxicity. The average optical density (O.D.) values of the blank control, CBS-400 extract, negative control and positive control are 1.10, 0.90, 1.04 and 0.49, respectively, as shown in Figure 7F. The cell viability value of CBS-400 group is 84% (>70%), indicating that CBS-400 does not have cytotoxic potential. The cell viability values of the negative control (Al_2_O_3_) and positive control (0.3% phenol) are 94% and 44% of the blank control, respectively, indicating that the test was valid. Raw data of optical density values are provided in Appendix A.

#### 3.6.2. Intracutaneous Reactivity Test

The observations of skin irritation at 0, 24, 48 and 72 h post intracutaneous injection showed no skin redness or swelling on either test sample-injected sites or vehicle-injected sites in any observation period, indicating that, at any observation time, the average reaction to CBS-400 extracts was the same as that of the blank control. As shown in Table 1, the final scores of both polar and non-polar CBS-400 extracts are zero (<1.0), indicating that CBS-400 does not give rise to any sign of skin irritation. The positive scores of the positive controls (3.9 for polar and 3.2 for non-polar) confirm the validity of the test (Table 2). Datasets of all scores according to Draize scale are provided in Appendix A and Appendix A.

#### 3.6.3. Skin Sensitization Test

As indicated in Table 3, the difference in body weight change between the CBS-400 extract group and the blank control group is insignificant (*p* > 0.05) within the whole experiment period. No significant difference (*p* > 0.05) in lymph nodes weight is seen between the mice applied with CBS-400 extract and the mice with the extract vehicle. Neither significant difference (*p* > 0.05) in labeling indices between the mice with the CBS-400 extract and the mice with the blank control is seen. The labeling index of the lymph nodes of the mice applied with the positive control group is significantly higher (*p* < 0.05) than that of the blank control. All these results conclude that CBS-400 with SI value of 0.6 (<1.6) is not considered to be a potential sensitizer. Raw data of body weights, lymph nodes weights and labeling index (O.D. values) are provided in Appendix A, Appendix A and Appendix A.

### 3.7. Animal Implantation Study and Resorption of CBS-400 Implant

Typical H&E-stained histological micrographs of CBS-400 implantation group at 4W, 8W and 12W post-operation are presented in Figure 8. At 4W, new bone is observed to extensively grow into the macro-pores of the implanted granules, while the struts of the porous granules are fragmented into pieces. At this stage, abundant new bone formation-involved osteoblasts and few bone-resorption-involved osteoclasts are seen throughout the sample. It is also observed that osteoclasts can penetrate into the intra-strut micro-pores of the implanted porous granules (Figure 8B). At 8W, extensive bone remodeling is ongoing, evidenced by the abundant presence of both osteoblasts and osteoclasts adjacent to the new bone and residual implant. At this stage, the new bone is remodeled to a trabecula-liked structure with more lamella structures, cement lines and lining cells. At 12W, the morphology of the new bone has become more like native bone, wherein fewer osteoblasts and osteoclasts are observed.

Typical TB-stained histological micrographs of the implantation group at 3D, 4W, 8W and 12W post-operation are presented in Figure 9. At 3D, the morphology of the porous implant is well maintained within the defect, and no bone regeneration is evidenced, due to its short implantation time. At 4W, in general, excellent bonding between the host bone and the implant is seen substantially without interposition of fibrous tissue. At this stage, numerous residual CBS-400 granules are observed to embed in the surrounding new bone which grow on the surfaces of the granules, as well as penetrate into the defect site, connecting to each other to form a dense new bone network with substantially random orientations. This new bone network becomes more porous at 8W, accompanied with a thickening of the trabecular bone and a decrease in the number and size of implant residues. With the bone remodeling process continuing at 12W, bone and implant residues both decrease, while many CBS-400 implant granules become too small to distinguish from the surrounding bone.

According to Materials and Methods 2.7, a 5 mm dia. circle (TZ) and its radially-reduced 4 mm dia. circle (IZ) were drawn on TB-stained superimposed composite images, and the measurements of residual implant and new bone within these two circles were compared. As shown in Figure 10, the number and size of implant residues both decrease with increasing implantation time. Comparison between TZ and IZ data indicates that the number and size of implant residues are not markedly different between these two zones. Statistical analyses of the residual implant and new bone obtained from TZ and IZ are presented in Figure 11. In the TZ zone, the average area ratio of residual CBS-400 decreases from 25.4% at 3D to 11.1% at 4W, 5.9% at 8W, and 3.8% at 12W post-operation. In other words, about 85% of the implant has readily been resorbed after 12 weeks. In the IZ zone, the average area ratio of residual CBS-400 decreases from 32.3% at 3D to 15.5% at 4W, 8.3% at 8W, and 5.1% at 12W, meaning that about 84% of the implant has been resorbed after 12 weeks. On the other hand, the average area ratio of new bone in the TZ zone decreases from 43.3% at 4W to 39.4% at 8W and 30.0% at 12W. The average area ratio of new bone in IZ zone decreases from 42.4% at 4W to 37.3% at 8W, and 30.4% at 12W. The new bone area at 3D was not measured, due to its substantial absence at this early stage. Datasets of residual implant ratio, new bone formation ratio, native bone area ratio at different time points within TZ and IZ are provided in Appendix A.

Linear regression of the residual implant ratio and quadratic regression of the new bone formation ratio versus logarithm of implantation period were performed, and the results are shown in Figure 12. Both regression analyses exhibit *p* < 0.001 for f-test, and all the regression coefficients and constants exhibit *p* < 0.001 for t-test. The correlation coefficients (R^2^) are 0.766 and 0.834, for bone formation ratio and residual implant ratio, respectively. The powers (1–β error probability) of both regressions are 99%. The correlation coefficients (about 0.8) seem to be high enough [22] to suggest that these analyses are convincing, and that the percentages of residual CBS-400 and new bone are quite predictable. According to the model, the expected implantation period for CBS-400 to be totally resorbed would be about 23W, and the bone area ratio would recover to the level of the blank at about 16W post-operation.

## 4. Discussion

All the major phases of the CBS-400 granules, including low-crystalline HA, CSD, DCPA and TTCP, are well-recognized as highly biocompatible and resorbable materials. Compared to the stoichiometric HA, the structure of the low-crystalline HA in CBS-400 is more toward the natural bone tissue [23,24] and could resorb faster in vivo [24,25]. TTCP and DCPA would eventually dissolve and form HA in a neutral pH environment. Since low-crystalline HA could have a wide Ca/P range from 1.2 to 2.2 [25,26], calculations of the element contents of CBS-400 by XRD-demonstrated phase contents based on a stoichiometric HA (Ca/P = 1.67) could lead to inaccurate results. Instead, the more time-consuming SEM/EDS technique was used for the determination of Ca/P and Ca/P/S ratios of the present CBS-400.

The CBS-400 granules were designed to have particle sizes ranging from about 400 μm to 1200 µm exhibit an irregular, rough and porous morphology, with numerous macro-pores and micro-pores. According to Hirschhorn et al. [27], the preferred minimum particle size would be about 400 µm for new bone formation. The selection of 1200 µm as an upper limit is due to the practical consideration that the particles may be delivered into bone cavities using a minimally invasive delivery tool, wherein particles that are too large would be difficult to deliver. After immersion in Hank’s solution for 7 days, the overall morphology, shape and integrity of the granules remain similar to that of the non-immersed samples, which is a good indication that the highly porous granules would probably survive the early implantation stage without a premature disintegration of the structure.

The pore sizes of CBS-400 are majorly distributed between about 60 µm and 180 µm. According to the animal studies of Hulbert et al. [28], Klawitter et al. [29] and Galois and Mainard [30], the minimum pore size for facilitating bone ingrowth would be about 75–100 µm. The far majority of the pores of CBS-400 have sizes larger than this, indicating that the porous morphology of the present granules would be able to facilitate new bone ingrowth. Furthermore, according to Bohner and Baumgart [31], the pore size between about 100 µm and 200 µm would be optimal to result in a proper specific surface area (SSA), which may help new bone ingrowth by increasing protein (such as vitronectin and fibronectin) adhesion to facilitate osteoblastic cell adhesion and spreading [32].

The micro-pores within the struts of CBS-400 might also promote new bone generation. The study of Coathup et al. [33] indicated that bone formation was evident within the strut, and greater bone formation was seen in scaffolds with increased strut porosity. In the present study, new bone formation was found within the struts of the implanted granules, indicating that the strut porosity could partially contribute to the observed fast resorption and new bone formation in CBS-400.

The early drop in pH of the Hank’s solution is considered to be a result of the dissolution of calcium sulfate in the presence of calcium phosphate to form octacalcium phosphate (OCP) and HA [34]. A number of studies have discussed the pH effects on bone healing/regeneration, but their results are not consistent. Walsh et al. [35] suggested that the local acidity and subsequent demineralization of adjacent bone and release of matrix-bound BMPs lead to a stimulatory effect on bone regeneration. Arnett [36] reported that acidosis exerts a reciprocal inhibitory effect on the mineralization of bone matrix by cultured osteoblasts. Shen et al. [37] found that, in a slightly basic microenvironment, the proliferation and alkaline phosphatase (ALP) activity of osteoblasts could be enhanced.

The large difference in 5-day P release between the extreme solution test and the simulation solution test (25 mg vs. 0.29 mg) could be explained by the high sensitivity of P release to the pH value of the solution. The pH values of the solutions used for the extreme solution and simulation solution tests are 3.0 and 7.4, respectively. According to Chow [38], the dissolution rates of calcium phosphates can be hundreds of times larger when the solution changes from neutral to acidic. The dissolution rates of calcium sulfates, on the other hand, are rather insensitive to the pH value of the solution [39].

The dissolution/solubility data may be helpful in assessing the safety of CBS-400 under clinical conditions. First of all, calcium, phosphorus and sulfur are all essential elements which are required at levels greater than 100 mg/day by adults [40]. Calcium ions are a major component for the FDA-approved infusions such as calcium gluconate and calcium chloride, to treat acute symptomatic hypocalcemia. In an early study, [41] used calcium infusions in the diagnosis of metabolic bone disease. An infusion of calcium gluconate (15 mg of calcium per kg of body weight) was given in 500 mL of physiological saline solution to 98 patients for 4 h. The results indicated that none of the patients showed any signs of calcium intoxication, and the serum calcium level had almost invariably returned to its original value after 20 h. The dosage used in this study was much higher than the Ca released from a 40 mL (10 g) grafted CBS-400 (the majority of clinical cases require less than 10 mL) for an average 70 kg person a day. In a relatively recent study of [42], intravenous calcium gluconate was given for 3 consecutive days to female patients undergoing assisted reproduction technique (ART) cycles for the prevention of ovarian hyperstimulation syndrome (OHSS). Such side effects as allergic reactions, anaphylaxis and symptoms of hypercalcemia were not observed in the study group. Again, the dosages of calcium used in these studies were twice higher than the mass of Ca released from 40 mL (10 g) grafted CBS-400 for an average person a day.

Most of the phosphorus in blood exists as phosphates or esters. It is known that phosphate is required for the generation of bony tissue and functions in the metabolism of glucose and lipids. The normal phosphorus levels in the blood of adults are 2.7–4.5 mg/dL [43], or 135–225 mg if the total volume of blood is 5 L. The released mass of phosphorus from CBS-400 for one person per day would be too small (<2 mg) to cause any significant fluctuation of normal phosphorus level. Even if all the P released from CBS-400 is taken by blood into the cardiovascular system, it is still far lower than the normal phosphorus level in the blood.

Sulfate ions exist in the magnesium sulfate infusion for replacement therapy in magnesium deficiency, especially in acute hypomagnesemia accompanied by signs of tetany similar to those observed in hypocalcemia. Van Norden et al. [44] used MgSO_4_ infusion to treat aneurysmal subarachnoid hemorrhage (SAH). With an intravenous dosage of 64 mmol magnesium sulfate a day for 14 days, no severe side effects were observed in the vast majority of patients with SAH. Magnesium sulfate injection is also used for the prevention and control of seizures in pre-eclampsia and eclampsia. The mass of sulfur element released from 40 mL (10 g) CBS-400 for an average person per day is about 90 mg, or 270 mg of sulfate ion, a level much lower than the dosages used for these indications.

Although the released calcium and sulfate ions could lower the surrounding pH value when immersed in aqueous solution, CBS-400 demonstrates good biocompatibility features, confirmed by the experimental results of cytotoxicity, intracutaneous reactivity and skin sensitization tests. The adherent and extended morphology of NIH/3T3 cells along with good cell viability found in the cytotoxicity test indicates that CBS-400 does not have a cytotoxic potential. In addition, CBS-400 does not induce skin irritation or skin sensitization that can stimulate the inflammation system and cause lymphocyte proliferation.

In the international standard ISO 10993-6, rabbit, femur bone and cylindrical-shaped defect are recommended as one of the appropriate species, implant site and defect shape respectively for the bone implantation test. Furthermore, according to the FDA 510(k) Special Guidance for Resorbable Calcium Salt Bone Void Filler Device, a critical size defect should be used. According to the studies of Prieto et al. [45] and Chen et al. [46], using the rabbit femur condyle model, the critical bone defect size would be about 5 mm in diameter × 8 mm in depth. The bone defect size used in the present study is 5 mm in diameter × 10 mm in depth. The large empty space appearing in the negative control (Appendix A in Appendix A) clearly indicates that the surgically created defect is large enough so that it cannot be repaired by normal bone healing mechanisms, consistent with the findings of Prieto et al. [45] and Chen et al. [46]

Except at the very early stage (3D), the residual implant and new bone ratios obtained from the TZ and IZ are not statistically different, indicating that implant resorption and new bone formation proceed quite uniformly throughout the entire implantation site, due to the highly porous feature of the implant. The similarity between the TZ and IZ data also indicates that the drawing of the original 5 mm circles was performed quite reliably. The somewhat lower residual implant ratio in TZ than IZ at 3D is not unexpected, due to the unavoidable presence of a thin gap between the implant and the void wall.

According to Bohner et al. [47], the resorption of CSD is mainly by physicochemical processes, specifically by dissolution. On the other hand, the resorption processes of calcium phosphates are apparently more complicated, which involve dissolution, phase transformation, osteoclast-mediated resorption and fragmentation, with particles being phagocytized by macrophages or engulfed by giant cells [48]. Once osteoclasts attach an implant surface, an intimate structure between osteoclasts and bone is formed, and a ‘sealing zone’ is developed. Within the sealing zone, demineralization of the implant material involving acidification of the isolated extracellular microenvironment takes place. Through a series of ion transport events, osteoclasts secrete HCl to form a low pH (~4.5), resorptive microenvironment to dissolve calcium phosphates. According to Xia and Triffitt [49], macrophages respond to small particles (typically <10 μm) by internalization, via phagocytosis and intracellular digestion. For larger particles, the macrophages fuse together, forming giant cells engulfing and digesting the particles, or through release of enzymes and/or lowering in pH for bulk digestion.

It is known that the resorption rate of HA is highly dependent on the crystallinity of HA. Klein et al. [50] reported that their highly crystalline HA did not show any degradation even at 9 months post-operation in a rabbit tibia model. However, a poor or nano-crystalline HA was found to be largely resorbed within 12 weeks post-operation in a beagle alveolar bone model [51] or NZW rabbit radial defect model [52].

CSD generally has a much higher degradation rate than HA. Liu et al. [53] investigated CSD containing 10% nano-crystalline HA in a NZW rabbit femur condyle model, and found that over 50% of the implant was resorbed after 4 weeks and almost totally resorbed within 8 weeks post-operation, yet with poor bone regeneration. In a relatively small NZW rabbit femur defect model (Ø 3.1 × 4 mm), Sheikh et al. [54] found that 40–50% DCPA resorbed with considerable hard callus formation within 4 weeks post-operation, and above 90% DCPA resorbed with apparent trabeculae formation after 12 weeks. Among the very few studies on the resorption rate of monolithic TTCP, it was observed that about 40% and 60% monolithic TTCP were resorbed in NZW rabbit condyle, at 4 and 24 weeks post-operation, respectively [55].

As mentioned earlier, the major phases of CBS-400 are low-crystalline HA and CSD with minor phases of DCPA and TTCP. According to the literature, a high resorption rate would be expected in CBS-400 during the early healing stage, whereas CSD in the material would quickly dissolve, followed by a relatively low resorption stage as calcium phosphates dominate. To a large extent, the observed resorption behavior of CBS-400 with the present animal model is consistent with the findings of the literature.

Little et al. [56] proposed that the bone healing process may be briefly divided into anabolic (new bone formation) and catabolic (bone resorption) responses, wherein the anabolic response dominates at the early stage, whereas the catabolic response dominates at the later stage. The combination of the two effects would logically lead to something like a Gaussian distribution in bone volume-bone healing time profile. In the present study, the maximum bone volume (anabolic response) is observed in the 4W samples, much faster than other studies using similar animal models [57,58].

Studies [59,60,61] using different calcium phosphate-based bone substitutes (beta-TCP, biphasic calcium phosphate (60% HA/40% TCP), deproteinized bovine bone, etc.) implanted in NZW rabbit femur condyle indicate that, at 6–8 weeks post implantation, the new bone ratios are 20–30%, while the residual implant ratios are 45–60%. Compared to these studies, the residual CBS-400 implant ratios (11–15% at 4W and 6–8% at 8W) are much lower, while the new bone ratios (42–43% at 4 weeks and 37–39% at 8 weeks) are much higher. The fast bone regeneration can help stabilize implant fixation, which is one key requirement to achieving fast and effective osteointegration [62]. The proper intergranular and intragranular spaces built in the CBS-400 implant are critical to the fast osteogenesis and resorption processes [63].

The regression line of residual implant versus natural log of implantation period indicates a decreasing resorption rate in vivo. This decrease may be attributed to the less surface area of CBS-400 granules exposed to cells with implantation time. The high osteoconductivity and fast bone formation observed in CBS-400 at 4W post-operation could be partly attributed to the numerous macro-pores, as well as the interconnectivity of these pores. In addition, the osteoclasts entering the struts through micro-pores could accelerate the cell-mediated resorption, thereby increasing interconnectivity for bone regeneration at an early healing stage.

From a biochemical point of view, calcium sulfate and HA, the two major components of CBS-400, play a major role in bone healing. According to Ricci et al. [64], calcium sulfate implanted in a bone defect can rapidly dissolve in body fluid and release calcium ions, which react with phosphate ions to form a layer of calcium phosphate, forming an osteoblast-friendly environment. According to Walsh et al. [35], the acidity caused by the dissolution of calcium sulfate and the precipitation of calcium phosphate may demineralize adjacent bone and release matrix bound BMPs, resulting in a stimulatory effect on bone regeneration. More recently, Aquino-Martínez et al. [65] reported that calcium sulfate could promote in vitro mesenchymal stem cell (MSC) migration and bone regeneration in vivo, by attracting the host’s osteoprogenitors into the implanted cell-free scaffold. Although calcium sulfate may dissolve too fast to allow bone to deposit on it, causing a non-continuous new bone formation in vivo [66], the porous CBS-400 maintains its structural integrity by the HA-dominated “scaffold” when its calcium sulfate is dissolved.

HA is considered to have the ability to directly bond to bone, possibly by natural bone cementing mechanisms [67]. This intimate contact has been observed throughout the entire implantation period in the present study. Bagambisa et al. [68] pointed out that, when the surface of HA is exposed to an aqueous environment, a process of elution and concomitant reprecipitation brings forth the formation of a transformational layer composed of spherocrystallites of colloidal dimensions. This elution/ re-precipitation layer resulting from the extensive degradation/ recrystallization events leads to a wide bonding layer for direct bone tissue deposition. Furthermore, the study of Chen et al. [69] indicated that calcium phosphate could attract and promote the differentiation of MSCs toward vascular endothelial cells to help the crucial revascularization process, and toward osteoblasts to regenerate new bone [70]. It seems that an appropriate combination of calcium sulfate and HA, such as CBS-400, may inherit the advantageous features of both components in implant resorption and bone regeneration processes. Furthermore, the soluble phases and highly porous morphology of CBS-400 could accelerate the process of biological apatite deposition and enhance bone-forming activity by providing more sites for cellular interaction. Other factors, such as pH and extracellular calcium ion concentration, could possibly also influence the cell-mediated resorption behavior of CBS-400 [71].

As a final remark, the H&E and TB-stained histological observations of the implanted CBS-400 and its surrounding bone morphology at 3D, 4W, 8W, and 12W post-implantation indicate that the implanted CBS-400 granules are intimately bonded to the surrounding new bone at all times. The measurements of residual implant material and the newly formed cancellous bone indicate that CBS-400 is not merely a degradable bone substitute. The resorption of CBS-400 appears simultaneously replaced by the formation of new cancellous bone. During the implantation time from 3D to 12W, the average residual CBS-400 ratio decreases with time from 25.4% down to 5.1%, meaning that about 85% of the implant has readily been resorbed after 12W. Within the same time frame, the newly formed cancellous bone ratio quickly increases to 42.4% at 4W (indicating a speedy new bone formation process at the early stage of implantation), followed by a bone remodeling process toward normal cancellous bone, wherein the new cancellous bone ratio gradually tapers down to 30.4% at 12W (the cancellous bone ratio in the normal condyle is 23.3%). These data show that the bone remodeling process toward normal cancellous bone is very close to completion at 12W. The detailed mechanisms behind the speedy new bone formation process at the crucial, early stage of implantation are apparently worth further investigation, including such topics as the early inflammation process, cells recruited for new tissue regeneration, bone mineralization process and neovascularization.

## 5. Conclusions

XRD patterns show that CBS-400 is comprised majorly of HA and CSD, with relatively small amounts of TTCP and DCPA. The SEM/EDS-determined Ca/P/S and Ca/P atomic ratios of the material are 54.61/39.21/6.18 and 1.39, respectively.Structural integrity test results show that, after immersion for 7 days, the overall morphology, shape and integrity of the Hank’s solution-immersed CBS-400 granules remain similar to that of non-immersed samples, without showing apparent collapse or disintegration under SEM.The average pH value of the Hank’s solution wherein CBS-400 is immersed for 1 day drops to 5.28. With immersion time, the pH value continues to increase to 6.55 after 7 days, and 7.08 after 14 days.In the extreme solution test, the average concentrations of Ca, P, and S in buffered citric acid solution are respectively 1208.1, 1285.4 and 570.9 ppm. In the simulation solution test, the concentrations of Ca, P, and S in TRIS-HCl are respectively 243.8, 4.6 and 443.6 ppm after immersion for 1 day; and respectively, 413.3, 14.5 and 670.2 ppm after 5 days.Cytotoxicity, intracutaneous reactivity and skin sensitization tests demonstrate the good biocompatibility features of CBS-400.The rabbit implantation results indicate that implanted CBS-400 granules are intimately bonded to the surrounding new bone at all times. The measurements of residual implant material and newly-formed cancellous bone reveal that the resorption of the implant is simultaneously replaced by the formation of new cancellous bone. During the implantation time from 3D to 12W, the average residual CBS-400 ratio decreases with time from 25.4% to 5.1%, meaning that about 85% of the implant has been resorbed after 12W. Within the same time frame, the newly formed cancellous bone ratio quickly increases to 42.4% at 4W, followed by a bone remodeling process toward normal cancellous bone, wherein the new cancellous bone ratio gradually tapers down to 30.4% after 12W.

## Figures and Tables

**Figure 1 materials-13-03458-f001:**
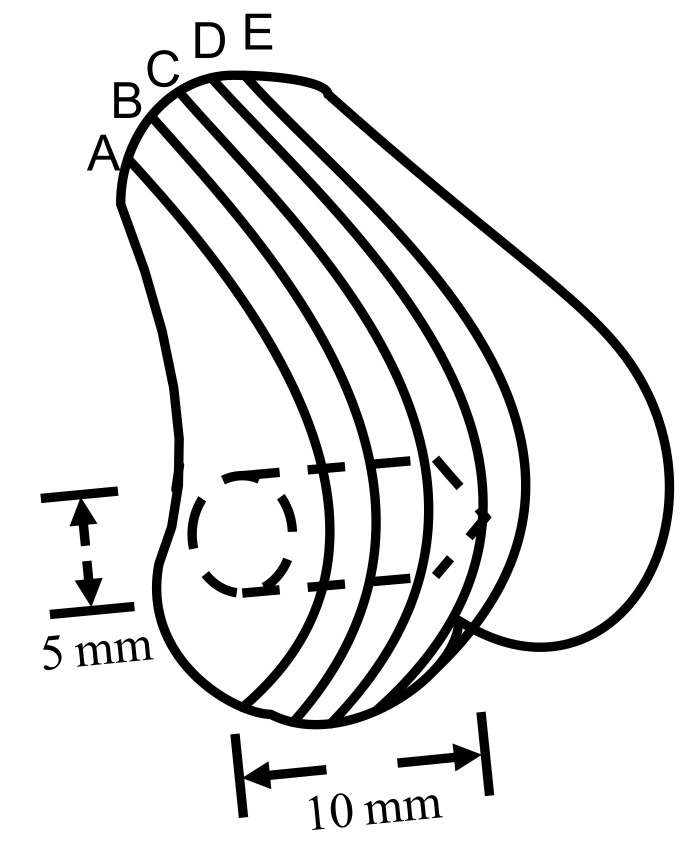
Illustration of 5 mm dia., 10 mm deep cylindrical-shaped CBS-400 implantation site. Lines A, B, C, D and E demonstrate the specific locations of the femur bone being sectioned for histological observation.

**Figure 2 materials-13-03458-f002:**
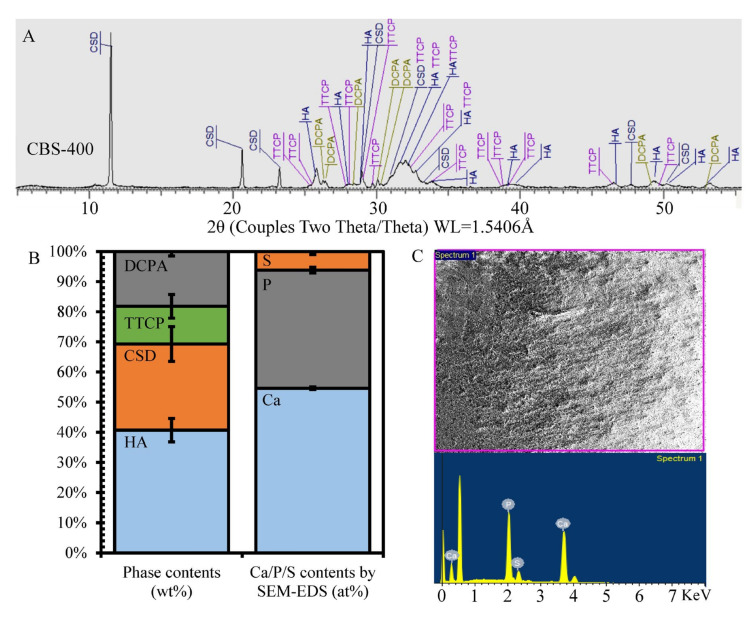
Phase identification and composition analysis of CBS-400. (**A**) XRD pattern; (**B**) XRD-determined phase contents and SEM/EDS-determined elemental contents; (**C**) SEM/EDS spectrum and analyzed region.

**Figure 3 materials-13-03458-f003:**
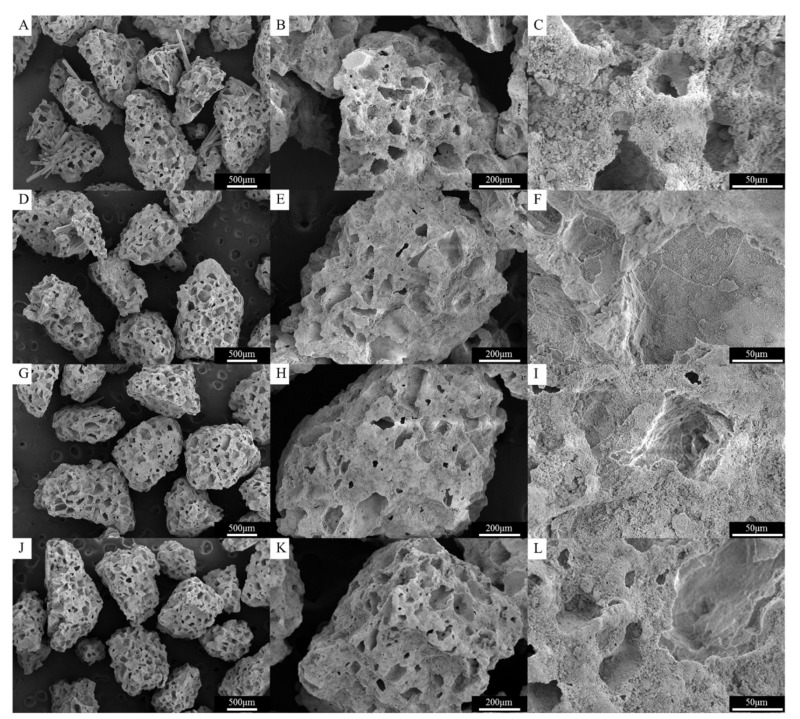
Surface morphology, shape and integrity of CBS-400 granules without immersion (**A**–**C**) and after immersion in Hank’s solution for 1 (**D**–**F**), 3 (**G**–**I**) and 7 (**J**–**L**) days.

**Figure 4 materials-13-03458-f004:**
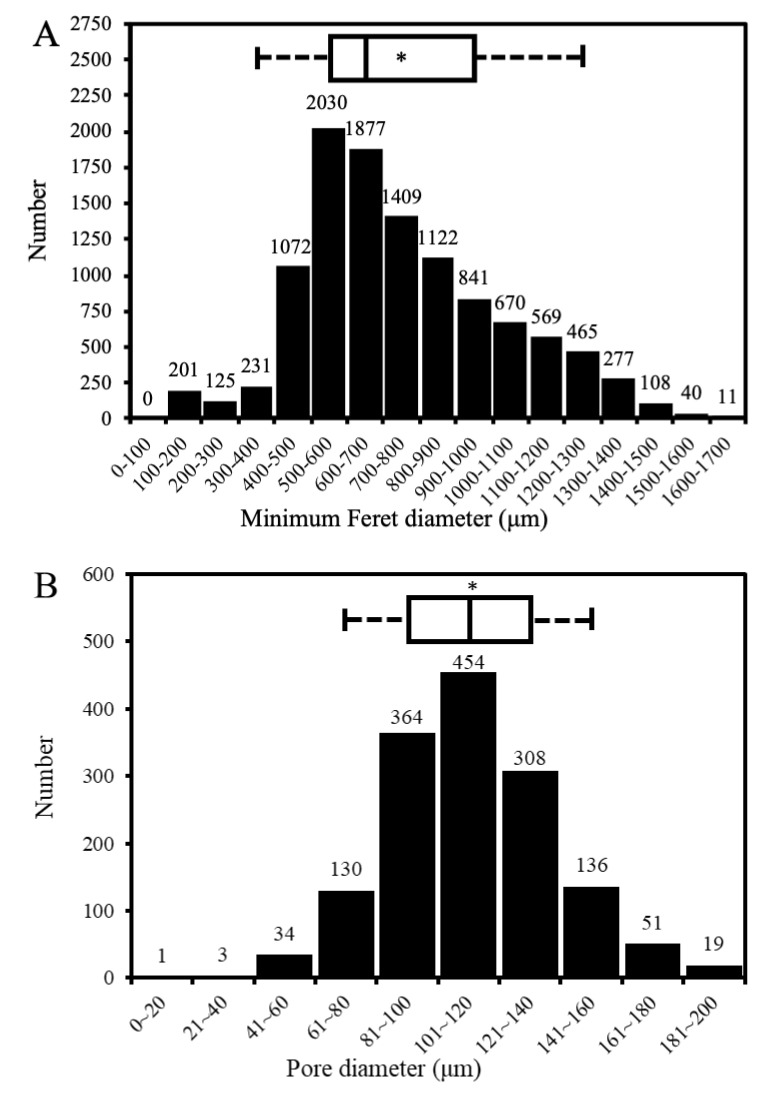
CBS-400 particle size distribution (**A**) and pore size distribution (**B**). Asterisk indicates the average number, and the two boundaries of the box plot above the bar graph define the 5th (left) and 95th (right) percentiles.

**Figure 5 materials-13-03458-f005:**
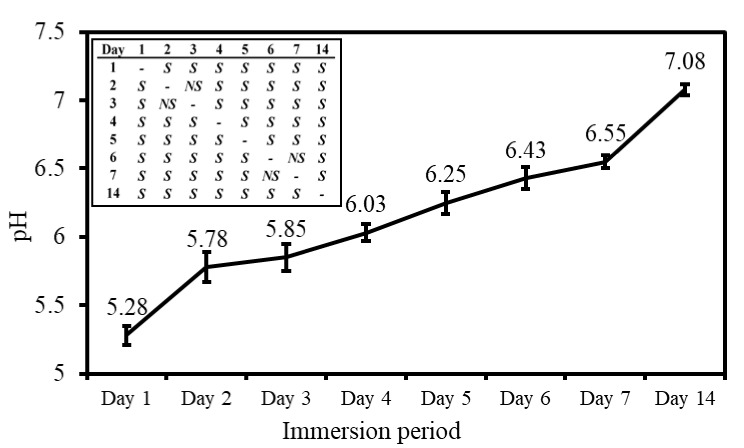
pH values in daily refreshed Hank’s solution, wherein CBS-400 is immersed. The significance of differences between two pH values at different immersion days are shown in the upper left table; S and NS represent *p* < 0.05 and *p* > 0.05, respectively.

**Figure 6 materials-13-03458-f006:**
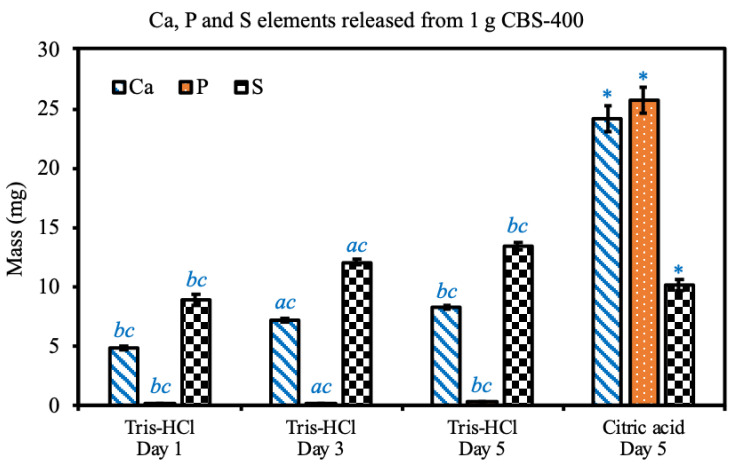
Ca, P and S elements released from 1 g CBS-400 in 20 mL TRIS-HCl at 37 °C for 1, 3, and 5 days; and in 20 mL buffered citric acid solution at 37 °C for 5 days. Symbols a, b and c indicate that the mean value has a significant difference (*p* < 0.05) compared to Day 1, Day 3 and Day 5 groups, respectively. Asterisks indicate that there are significant differences between TRIS-HCl and citric acid groups at day 5.

**Figure 7 materials-13-03458-f007:**
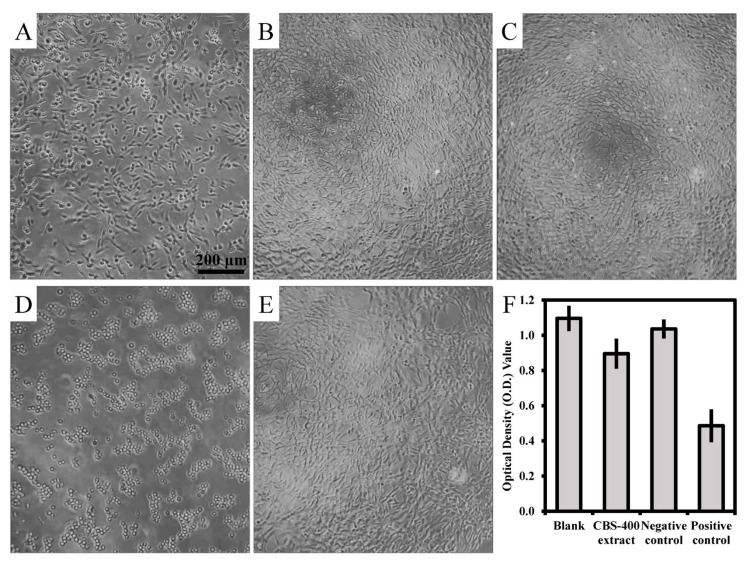
Morphologies of NIH/3T3 cells before treatment (**A**), treated with culture medium as blank (**B**), treated with CBS-400 extract (**C**), treated with 0.3% phenol as positive control (**D**), treated with Al_2_O_3_ extract as negative control (**E**), and their O.D. values (**F**).

**Figure 8 materials-13-03458-f008:**
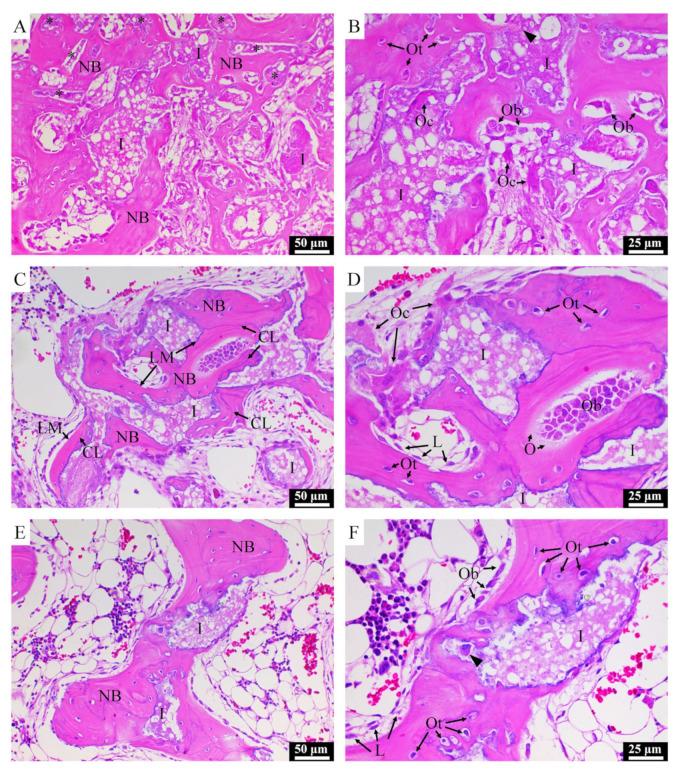
H&E-stained histological images of NZW rabbit femur condyle of CBS-400-implanted group at 4W (**A**,**B**), 8W (**C**,**D**) and 12W (**E**,**F**) post-operation. I: residual implant; NB: new bone; LM: lamellar matrix; CL: cement line; Ob: osteoblast; Oc: osteoclast; Ot: osteocyte; O: osteoid; L: lining cell. Asterisks indicate implant residues embedded in surrounding new bone. Arrow heads indicate new bone grown in implant micro-pores.

**Figure 9 materials-13-03458-f009:**
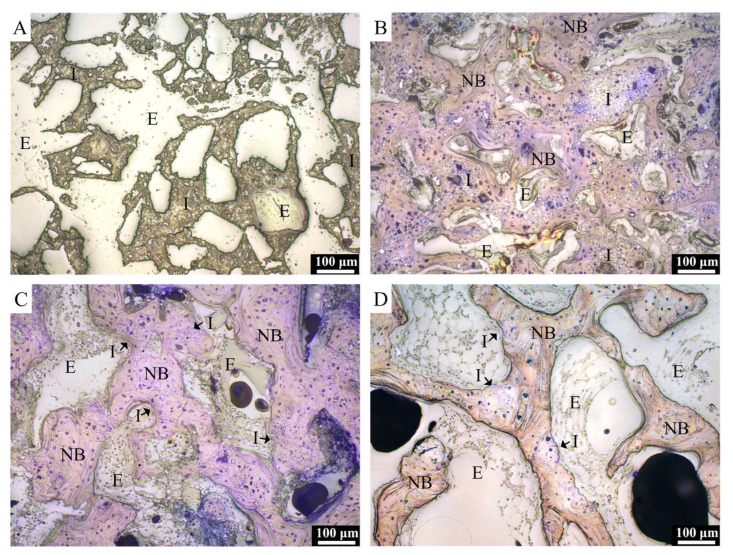
TB-stained histological images of NZW rabbit femur condyle in CBS-400-implanted group at 3D (**A**), 4W (**B**), 8W (**C**) and 12W (**D**) post-operation. I: residual implant; NB: new bone; E: epoxy.

**Figure 10 materials-13-03458-f010:**
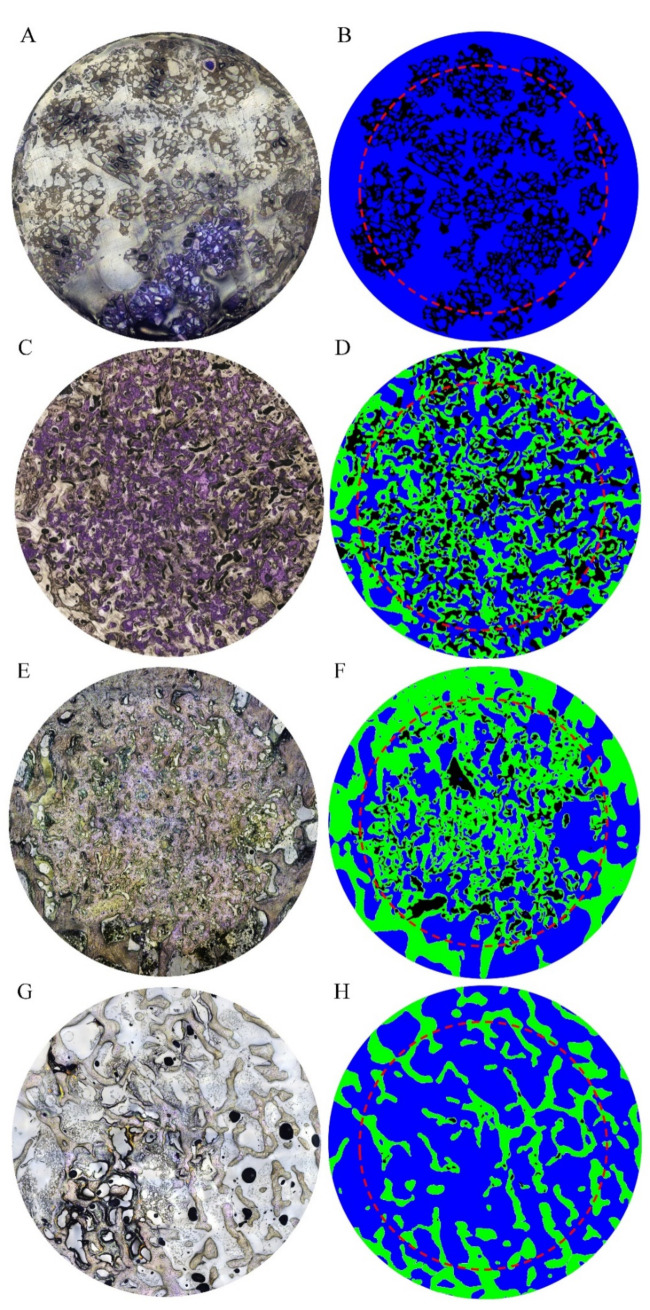
Overall, 5 mm dia. TB-stained superimposed composite images of NZW rabbit femur condyle in CBS-400-implanted group at 3D (**A**), 4W (**C**), 8W (**E**) and 12W (**G**) post-operation and their corresponding manually drawn images (**B**,**D**,**F** and **H**), illustrating residual CBS-400 (black), new bone (green) and marrow space (blue). Moreover, 4 mm dia. inner zones are outlined by red dash circles.

**Figure 11 materials-13-03458-f011:**
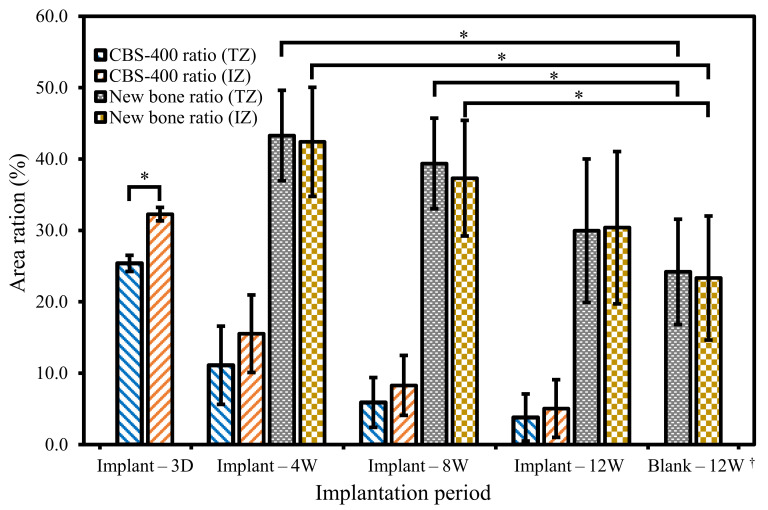
Area ratios of residual CBS-400 implant and new bone at 3D, 4W, 8W and 12W post-operation. The area ratios of native bone in blank group at 12W are also included. Data from TZ and IZ zones are both presented for comparison. * Significant (*p* < 0.05). ^†^ Native bone area ratio instead of new bone area ratio.

**Figure 12 materials-13-03458-f012:**
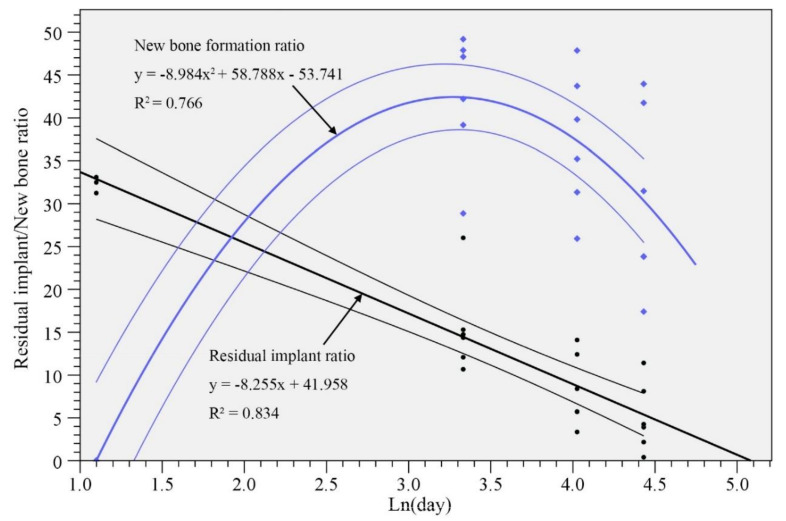
Residual implant and new bone ratios. Linear regression (black lines) and quadratic regression (blue lines) are applied to new bone ratio and residual implant ratio, respectively. The regression lines and their 95% confidence intervals are drawn.

**Table 1 materials-13-03458-t001:** Scores of CBS-400 extract and blank control sites for intracutaneous (intradermal) reactions (Draize scale).

**Score of Polar Solvent Extraction Group**							
		Animal No.1		Animal No.2		Animal No.3	
		Extract	Vehicle	Extract	Vehicle	Extract	Vehicle
Im.	Erythema	0 ^1^	0	0	0	0	0
	Oedema	0	0	0	0	0	0
24 h	Erythema	0	0	0	0	0	0
	Oedema	0	0	0	0	0	0
48 h	Erythema	0	0	0	0	0	0
	Oedema	0	0	0	0	0	0
72 h	Erythema	0	0	0	0	0	0
	Oedema	0	0	0	0	0	0
Avg. of 3 time points (erythema + oedema)		0	0	0	0	0	0
Avg. of all extract sites (A) = 0; Avg. of all vehicle sites (B) = 0							
Final score of irritation of CBS-400 extract (A−B)					0		
**Score of Non-Polar Solvent Extraction Group**							
		Animal No.1		Animal No.2		Animal No.3	
		Extract	Vehicle	Extract	Vehicle	Extract	Vehicle
Im.	Erythema	0	0	0	0	0	0
	Oedema	0	0	0	0	0	0
24 h	Erythema	0	0	0	0	0	0
	Oedema	0	0	0	0	0	0
48 h	Erythema	0	0	0	0	0	0
	Oedema	0	0	0	0	0	0
72 h	Erythema	0	0	0	0	0	0
	Oedema	0	0	0	0	0	0
Avg. of 3 time points (erythema + oedema)		0	0	0	0	0	0
Avg. of all extract sites (A) = 0; Avg. of all vehicle sites (B) = 0							
Final score of irritation of CBS-400 extract (A−B)					0		

^1^ Average score of 5 injection sites. Im.: Observe after injection immediately; Avg.: average.

**Table 2 materials-13-03458-t002:** Scores of positive control and blank control sites for intracutaneous (intradermal) reactions (Draize scale).

**Score of Polar Solvent Positive Control Group**							
		Animal No.1		Animal No.2		Animal No.3	
		Formalin	Vehicle	Formalin	Vehicle	Formalin	Vehicle
Im.	Erythema	0 ^1^	0	0	0	0	0
	Oedema	0	0	0	0	0	0
24 h	Erythema	0.4	0	2.0	0	1.0	0
	Oedema	0.6	0	3.0	0	2.0	0
48 h	Erythema	1.2	0	2.2	0	1.4	0
	Oedema	1.6	0	3.0	0	3.0	0
72 h	Erythema	1.8	0	2.8	0	2.4	0
	Oedema	1.2	0	2.4	0	3.0	0
Avg. of 3 time points (erythema + oedema)		2.3	0	5.1	0	4.3	0
Avg. of all positive control sites (A) = 3.9; Avg. of all vehicle sites (B) = 0							
Final score of irritation of formalin (A−B)					3.9		
**Score of Non-Polar Solvent Positive Control Group**							
		Animal No.1		Animal No.2		Animal No.3	
		Histamine	Vehicle	Histamine	Vehicle	Histamine	Vehicle
Im.	Erythema	0	0	0	0	0	0
	Oedema	0	0	0	0	0	0
24 h	Erythema	0.6	0	0.6	0	1.8	0
	Oedema	1.4	0	0.6	0	2.4	0
48 h	Erythema	2.0	0	1.0	0	1.4	0
	Oedema	1.4	0	1.4	0	2.6	0
72 h	Erythema	2.2	0	1.4	0	1.8	0
	Oedema	1.4	0	1.6	0	3.0	0
Avg. of 3 time points (erythema + oedema)		3.0	0	2.2	0	4.3	0
Avg. of all positive control sites (A) = 3.2; Avg. of all vehicle sites (B) = 0							
Final score of irritation of histamine (A−B)					3.2		

^1^ Average score of 5 injection sites. Im.: Observe after injection immediately; Avg.: average.

**Table 3 materials-13-03458-t003:** Body weights and weights of lymph nodes, BrdU labeling index and SI values of the mice for LLNA:BrdU-ELISA test.

		CBS-400 Extract Group	Vehicle Control (AOO) Group	Positive Control (DNCB) Group
Body weight (g)	day 0	20.2 ± 0.7	20.5 ± 1.2	19.6 ± 1.3
	day 1	20.9 ± 0.5	20.6 ± 1.1	20.5 ± 1.2
	day 2	20.3 ± 1.1	20.8 ± 1.2	20.2 ± 0.9
	day 3	20.3 ± 0.8	21.0 ± 1.0	20.2 ± 1.0
	day 4	20.3 ± 0.9	20.9 ± 1.2	19.5 ± 0.9
	day 5	20.0 ± 0.8	20.7 ± 1.0	19.8 ± 0.4
Lymph nodes weight (mg)		5.1 ± 1.3	4.7 ± 0.7	22.0 ± 2.5 *
Labeling index		0.100 ± 0.056	0.167 ± 0.026	0.431 ± 0.050 *
Stimulation Index		0.60	1	2.58

*The value has significant difference compared to the vehicle control group (*p* < 0.05).

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
