# Peer review of "Physical/Chemical Properties and Resorption Behavior of a Newly Developed Ca/P/S-Based Bone Substitute Material"

_materials, 2020, doi:10.3390/ma13163458_

Round 1

Reviewer 1 Report

  1. The authors should provide results or literature survey on the topic of the biodegradation rate of the materials studied with time. Does the materials composition fit the tissue formation rate at a given materials characteristics.
  2. As far as I understand, the granules can be used in conditions without load bearing. Please discuss the size of the bone defects can be treated with these granules in more details.
  3. Can still the size of the granules and their porosity be optimized to better suit the purposes of bone tissue regeneration.

Author Response

A1.The authors should provide results or literature survey on the topic of the biodegradation rate of the materials studied with time. Does the materials composition fit the tissue formation rate at a given materials characteristics.

Q1. As suggested, a more detailed literature survey of the biodegradation rate of the studied materials, as well as a discussion of the tissue formation rates of CBS-400, were added in the revised manuscript. The following sentences were added in Discussion (pages 23-24, lines 627-646 in rev ms):

“It is known that the resorption rate of HA is highly dependent on the crystallinity of the HA. Klein et al. [50] reported that their highly crystalline HA did not show any degradation even at 9 months post-operation in a rabbit tibia model. However, a poor or nano-crystalline HA was found to be largely resorbed within 12 weeks post-operation in a beagle alveolar bone model [51] or NZW rabbit radial defect model [52].

CSD generally has a much higher degradation rate than HA. Liu et al. [53] investigated CSD containing 10% nano-crystalline HA in a NZW rabbit femur condyle model and found that over 50% of the implant was resorbed after 4 weeks and almost totally resorbed within 8 weeks post-operation, yet with poor bone regeneration. In a relatively small NZW rabbit femur defect model (Ø 3.1 x 4 mm), Sheikh et al. [54] found that 40-50% DCPA resorbed with considerable hard callus formation within 4 weeks post-operation, and above 90% DCPA resorbed with apparent trabeculae formation after 12 weeks. Among the very few studies on the resorption rate of monolithic TTCP, it was observed that about 40% and 60% monolithic TTCP were resorbed in NZW rabbit condyle at 4 and 24 weeks post-operation, respectively [55].
As mentioned earlier, the major phases of CBS-400 are low-crystalline HA and CSD with minor phases of DCPA and TTCP. According to the literature, a high resorption rate would be expected in CBS-400 during the early healing stage, whereas CSD in the material would quickly dissolve, followed by a relatively low resorption stage as calcium phosphates dominate. To a large extent, the observed resorption behavior of CBS-400 with the present animal model is consistent with the findings of the literature.”

Q2. As far as I understand, the granules can be used in conditions without load bearing. Please discuss the size of the bone defects can be treated with these granules in more details.

A2. We agree with the Reviewer that the porous granules cannot be used under a high load bearing condition.

Other animal models with different bone defect sizes were conducted to assess the clinical potential of CBS-400 in stimulating bone healing, for example, a bone defect of 2 mm in diameter and 3 mm in depth was used in a rat femoral shaft model and a 8 mm in diameter and 10 mm in depth bone defect was used in a goat model. Since the results were not yet published, they were not referenced in this manuscript.

Q3. Can still the size of the granules and their porosity be optimized to better suit the purposes of bone tissue regeneration.

A3. Different sizes of granules and pores had been tested to optimize the performance of the implant. The presently-used granule and pore sizes had been optimized to our best efforts. These parameters, of course, may be further fine-tuned to suit certain purposes, e.g., smaller sizes for small dental bone defects and larger sizes for large scale defects (tumor removal-caused large bone defects, etc.).

Reviewer 2 Report

Was any reference for the animal model? The size and the shape of defects in this model should be validated or have a reference.
Figure 7 should have a scale bar.
In this study, authors claimed their materials were biodegradable. However, the mechanism for its degradation seemed to be unclear.
In Fig.9, 3D samples should have full of graft because it is too early for degradation. However, the space occupied by implants seemed to be small.
In Fig. 9, the space occupied by new bone seemed to be gradullay decreased from 4 w to 12 w. Why the implant was disappeared with new bone from 4 w to 12 w?

Author Response

Q1. Was any reference for the animal model? The size and the shape of defects in this model should be validated or have a reference.

A1. As suggested, 2 references (Refs 45 and 46 in rev ms) to the animal model were added.
The following sentences were added in Discussion (page 23, lines 597-606):

“In the international standard ISO 10993-6, rabbit, femur bone and cylindrical-shaped defect are recommended as one of the appropriate species, implant site and defect shape respectively for bone implantation test. Furthermore, according to the FDA 510(k) Special Guidance for Resorbable Calcium Salt Bone Void Filler Device, a critical size defect should be used. According to the studies of Prieto et al. [45] and Chen et al. [46] using rabbit femur condyle model, the critical bone defect size would be about 5 mm in diameter x 8 mm in depth mm. The bone defect size used in the present study is 5 mm in diameter x 10 mm in depth. The large empty space appearing in the negative control (Figure S1 in Supplementary Materials) clearly indicates that the surgically-created defect is large enough so that it cannot be repaired by normal bone healing mechanisms, consistent with the findings of Prieto et al. [45] and Chen et al. [46].”

Q2. Figure 7 should have a scale bar.

A2. As suggested,a scale bar was added in the figure (Figure 7 in rev ms).

Q3. In this study, authors claimed their materials were biodegradable. However, the mechanism for its degradation seemed to be unclear.

A3. As suggested,the following sentences were added in Discussion (page 23, lines 614-626 in rev ms):

According to Bohner et al. [47], the resorption of CSD is mainly by physicochemical processes, specifically by dissolution. On the other hand, the resorption processes of calcium phosphates are apparently more complicated, which involve dissolution, phase transformation, osteoclast-mediated resorption and fragmentation with particles being phagocytized by macrophages or engulfed by giant cells [48]. Once osteoclasts attach an implant surface, an intimate structure between osteoclasts and bone is formed and a ‘sealing zone’ is developed. Within the sealing zone, demineralization of the implant material involving acidification of the isolated extracellular microenvironment takes place. Through a series of ion transport events, osteoclasts secrete HCl to form a low pH (~4.5), resorptive microenvironment to dissolve calcium phosphates. According to Xia and Triffitt [49], macrophages respond to small particles (typically <10 μm) by internalization via phagocytosis and intracellular digestion. For larger particles, the macrophages fuse together forming giant cells engulfing and digesting the particles, or through release of enzymes and/or lowering in pH for bulk digestion.

Q4. In Fig.9, 3D samples should have full of graft because it is too early for degradation. However, the space occupied by implants seemed to be small.

A4. It should be reminded that the area of Fig. 9A could cover only one granule. Due to the highly porous morphology of the granules and the presence of inter-granular voids, the space literally occupied by the implant could be seemingly small under this magnification.

Q5. In Fig. 9, the space occupied by new bone seemed to be gradually decreased from 4 w to 12 w. Why the implant was disappeared with new bone from 4 w to 12 w?

A5. As described in Discussion (page 24, lines 639-646 in original ms), the resorption of CBS-400 and new bone formation proceed simultaneously but at differences paces. From 3D to 12W, the residual CBS-400 continues to decrease with time from 25.4% to 5.1%. Within the same time frame, the new bone ratio quickly increases to 42.4% at 4W, followed by a bone remodeling process toward normal cancellous bone, wherein the new bone ratio gradually tapers down to 30.4% at 12W. Therefore, the observation of decreases in both implant and new bone after 4W is not unexpected.

Reviewer 3 Report

July, 20th 2020

The authors investigate a series of physical/chemical properties, biocompatibility and the behavior of implant resorption, as well as new bone formation, of a newly-developed Ca/P/S-based bone substitute material (Ezechbone® Granule CBS-400). Many experimental tests performed (chemical/physical characterization, biocompatibility, etc.) demonstrate that: it has good biocompatibility features; its resorption and the formation of new cancellous bone proceed at the substantially same pace of the host bone; after implantation for 12 weeks, about 85% of the implant has been resorbed. The newly-formed cancellous bone ratio quickly increases to >40% at 4 weeks, followed by a bone remodeling process toward normal cancellous bone.

The work is interesting, well organized and clearly written, but some modifications need to improve the paper.

Major comments

The authors refer in Results:

  • (lines 337-338): … the surfaces of the granules and the edges around the pores become slightly smoother after immersion.

No major differences are appreciable; in addition to porosity tests, it would have been appropriate to perform roughness tests.

  • (lines 344-345)-Fig. 4A, (lines 360-361)-Fig. 5, (lines 376)-Fig. 6: these results don't have the statistics and that lowers their value a bit.

The authors refer in Discussion:

  • (lines 614-615): Aquino-Martínez et al. [54] reported that calcium sulfate could promote in vitro mesenchymal stem cell (MSC) migration and bone regeneration in vivo by attracting the host’s osteoprogenitors into the implanted cell-free scaffold.
  • (lines 626-628): Furthermore, the study of Chen et al. [58] indicated that calcium phosphate could attract and promote differentiation of MSCs toward vascular endothelial cells to help the crucial revascularization process, and toward osteoblasts to regenerate new bone.

Is there any observation that the authors made about the effect of the bone substitute tested on blood vessels (growth, permeability) in vivo or on endothelial cell proliferation in vitro.

The authors summarize the results (by points) in their Conclusions (lines 648-672): It would be more appropriate to emphasize the salient features of the bone substitute studied and place them in relation to future developments.

Minor comments

Line 391- Fig. 7: in the figure, scale the bars are missing.

Line 438 – Fig. 8: OB: osteoclast (no). replace with “osteoblasts”

Line 634: behavior of CBS-400 [0]: check the reference.

Overall and final opinion of the Reviewer:

The manuscript is well organized and the background and materials and methods are properly explained. The discussion is also comprehensive of the data obtained as well as aspects that have emerged from the literature. Less incisive appear the conclusions, which have been drafted as a list of results.

Some modifications/integration are required.

The manuscript is acceptable after a minor revision.

Author Response

Reviewer 3

Major comments

Q1. The authors refer in Results:
(lines 337-338): … the surfaces of the granules and the edges around the pores become slightly smoother after immersion.
No major differences are appreciable; in addition to porosity tests, it would have been appropriate to perform roughness tests.

A1. For small, irregular-shaped and highly porous particles, an accurate quantitative measurement of surface roughness is very difficult to conduct. To avoid confusion, if agreeable, we would like to remove the controversial sentence mentioned by the Reviewer, “As shown in the figures, the surfaces of the granules and the edges around the pores become slightly smoother after immersion.” (page 8, lines 337-338 in original ms).

Q2. (lines 344-345)-Fig. 4A, (lines 360-361)-Fig. 5, (lines 376)-Fig. 6: these results don't have the statistics and that lowers their value a bit.

A2. As suggested, statistical analyses were added to these figures (figure 4, 5 and 6 in rev ms). The following contents were added/modified accordingly:

  • Two sentences “Box plots were used to demonstrate the 5th percentile, 1st quartile (25th percentile), median (50th percentile), 3rd quartile (75th percentile) and 95th percentile to show the data distribution. One-way ANOVA was used to evaluate the difference in pH values in Hank’s solution and the masses of Ca, P and S elements released from CBS-400 in TRIS-HCl group between different immersion days.” were added in the Materials and Methods (page 7, lines 305-309 in rev ms).
    The original sentence “Unpaired Student’s t-test was also used to analyze the differences in residual implant and new bone ratios between the two circles, as well as the differences in bone area ratio between blank control and implantation groups.” (page 7, lines 306-308 in original ms) was modified as “Unpaired Student’s t-test was also used to analyze the differences in the masses of Ca, P and S elements released from CBS-400 between TRIS-HCl group and citric acid group, residual implant and new bone ratios between the two circles, as well as the differences in bone area ratio between blank control and implantation groups.” (page 7, lines 311-314 in rev ms).
  • A sentence “Asterisk indicates the average number and the two boundaries of the box plot above the bar graph define the 5th (left) and 95th (right) percentiles.was added in the caption of Figure 4 (page 10, lines 353-355 in rev ms).
  • The original sentence “The measurements indicate that less than 3% of the particles are smaller than 300 μm and about 87% of the particles are between 400 to 1200â€¯μ” (page 8, lines 344-345 in original ms) was modified as “The measurements indicate that less than 5% of the particles are smaller than 300 μm and about 90% of the particles are between 300 to 1300 μm, close to our designation.” (page 8, lines 340-342 in rev ms).
  • The original sentence “The distribution data indicate that 98-99% of the measured pores of the granules are between 40 μm and 180 μm; and about 89% between 80 μm and 200â€¯μ” (page 10, lines 351-352 in original ms) was modified to “The distribution data indicate that 98-99% of the measured pores of the granules are between 40 μm and 180 μm; and about 90 % between 60 μm and 160 μm.” (page 10, lines 361-362 in rev ms).
  • A sentence “The significance of differences between two pH values at different immersion days are shown in the upper left table; S and NS represent p<0.05 and p>0.05, respectively.” was added in the caption of Figure 5 (page 11, lines 373-375 in rev ms).
  • The original sentence “With increasing immersion time, the pH value continues to increase with immersion time.” (page 11, lines 359-360 in original ms) was modified as “With increasing immersion time, the pH value continues to increase significantly with immersion time.” (page 11, lines 369-371 in rev ms).
  • A sentence “Symbols a, b and c indicate that the mean value has a significant difference (p<0.05) compared to Day 1, Day 3 and Day 5 groups, respectively. Asterisks indicate that there are significant differences between TRIS-HCl and citric acid groups at day 5.” was added in the caption of Figure 6 (page 12, lines 393-396 in rev ms).
  • A sentence “At day 5, compared to the TRIS-HCl group, the citric acid group shows significant differences in Ca, P and S elements released from CBS-400. Within the TRIS-HCl group, the released Ca, P and S elements significantly increase with immersion time.” was added in the Results (page 11, lines 388-390 in rev ms).

Q3. The authors refer in Discussion:
(lines 614-615): Aquino-Martínez et al. [54] reported that calcium sulfate could promote in vitro mesenchymal stem cell (MSC) migration and bone regeneration in vivo by attracting the host’s osteoprogenitors into the implanted cell-free scaffold.
(lines 626-628): Furthermore, the study of Chen et al. [58] indicated that calcium phosphate could attract and promote differentiation of MSCs toward vascular endothelial cells to help the crucial revascularization process, and toward osteoblasts to regenerate new bone.

Is there any observation that the authors made about the effect of the bone substitute tested on blood vessels (growth, permeability) in vivo or on endothelial cell proliferation in vitro.

A3. Investigation of the effect of the bone substitute on blood vessel growth and permeability in vivo or on endothelial cell proliferation in vitro is beyond the focus of the present study and was not conducted. We appreciate the reviewer’s insightful suggestions which would be very helpful for our follow-up studies.

Q4. The authors summarize the results (by points) in their Conclusions (lines 648-672): It would be more appropriate to emphasize the salient features of the bone substitute studied and place them in relation to future developments.

A4. As suggested, the following sentences were added in Discussion (page 25, lines 708-711 in rev ms):

“The detailed mechanisms behind the speedy new bone formation process at the crucial, early stage of implantation apparently worth further investigation, including such topics as the early inflammation process, cells recruited for new tissue regeneration, bone mineralization process and neovascularization.”

Minor comments

Q5. Line 391- Fig. 7: in the figure, scale the bars are missing.

A5. As suggested, a scale bar was added in Figure 7 (Figure 7 in rev ms).

Q6. Line 438 – Fig. 8: OB: osteoclast(no). replace with “osteoblasts”.

A6. As suggested, “osteoclast” in Figure 8 was replaced with “osteoblast” (Figure 8 in rev ms).

Q7. Line 634: behavior of CBS-400 [0]: check the reference.

A7. As suggested, the reference in original ms (page24, line 634) was corrected (page 25, lines 695 in rev ms).

Round 2

Reviewer 2 Report

Authors addressed all issues raised by reviewer. After checking minor spelling errors, this manuscript can be published as it is.